# Characteristics of *Streptococcus agalactiae* Colonizing Nonpregnant Adults Support the Opportunistic Nature of Invasive Infections

Elisabete R. Martins,[a] Dulce Nascimento do Ó,[b] Ana Luísa Marques Costa,[b] José Melo-Cristino,[a] Mário Ramirez[a]

aInstituto de Microbiologia, Instituto de Medicina Molecular, Faculdade de Medicina, Universidade de Lisboa, Lisbon, Portugal
bAPDP—Diabetes Portugal, Lisbon, Portugal

**ABSTRACT** The prevalence and lineages of *Streptococcus agalactiae* (group B streptococci [GBS]) colonizing pregnant women are well studied, but less is known about colonization of nonpregnant adults. We characterized GBS colonization in adults as a potential reservoir for infections and tested for the presence of clones with a potentially higher invasive disease potential. We evaluated GBS gastrointestinal, genitourinary, and oral colonization among 336 nonpregnant adults in the community. We characterized the isolates by serotyping, multilocus sequence typing, profiling of surface protein genes and pili, and antimicrobial susceptibility and compared them with contemporary invasive isolates. The colonization rate ($n = 107$, 32%) among nonpregnant adults was like that of pregnant women. Colonization increased with age ($\sim$25% in the 18 to 29 and 30 to 44 years old groups and >42% in the $\geq$60 years old group), potentially explaining the higher incidence of disease with older age. Participants who were colonized at multiple sites (73%) were frequently carrying indistinguishable strains (93%), consistent with the existence of a single reservoir of colonization and transfer of GBS between sites within the same individual. The most frequent lineages found were serotype Ib/CC1 ($n = 27$), serotype V/CC1 ($n = 19$), serotype Ia/CC23 ($n = 13$), serotype III/ST17 ($n = 13$), and serotype Ib/CC10 ($n = 11$). Comparison with contemporary isolates causing invasive infections in Portugal did not reveal any lineage associated with either asymptomatic carriage or invasive disease. Asymptomatic colonization of nonpregnant adults is significant and could act as a reservoir for invasive disease, but in contrast to infant disease, we found no GBS lineages with an enhanced potential for causing invasive disease in adults.

**IMPORTANCE** The increasing incidence of *Streptococcus agalactiae* (group B streptococci [GBS]) infections in adults and the inability of antimicrobial prophylaxis peripartum to control late-onset infections in infants motivate the study of the asymptomatic carrier state in nonpregnant adults. We found an overall carriage rate like that of pregnant women, increasing with age, potentially contributing to the higher incidence of GBS infections with age. Colonization of diabetic participants was not higher despite the higher number of infections in this group. Comparison between contemporary genetic lineages causing infections and found in asymptomatic carriers did not identify particularly virulent lineages. This means that any prophylactic approaches targeting colonization by particular lineages are expected to have a limited impact on GBS disease in adults.

**KEYWORDS** group B *Streptococcus*, *Streptococcus agalactiae*, antimicrobial resistance, carriage, case-carrier ratio, clones, colonization, invasive disease, multilocus sequence typing, nonpregnant adults, serotype

Despite its pathogenic potential, *Streptococcus agalactiae* (or group B streptococci [GBS]) can frequently be found as part of the commensal microbiota (1, 2). Colonization of the vaginal mucosa of pregnant women is well established, and passage through the birth canal

Address correspondence to Mário Ramirez, ramirez@fm.ul.pt.

The authors declare a conflict of interest. J.M.-C. has received research grants administered through his university and received honoraria for serving on the speakers' bureau of Pfizer and Merck Sharp and Dohme. M.R. has received honoraria for serving on the speakers' bureau of Pfizer and Merck Sharp and Dohme and for serving in specialist panels of GlaxoSmithKline and Merck Sharp and Dohme. The funders had no role in the design of the study; in the collection, analyses, or interpretation of data; in the writing of the manuscript, or in the decision to publish the results.

is thought to be the major source of GBS in early-onset disease (1). The source of GBS in late-onset neonatal disease is less clear, and horizontal transmission from caregivers is thought to be involved. The most frequently colonized site is the gastrointestinal tract, which is frequently assumed to be the natural reservoir of GBS, as well as the likely source of vaginal and oral colonization (3).

In European countries, the prevalence of GBS carriage among pregnant women ranges from 6.5% to 36%, with most countries reporting colonization rates within 15 to 20% (4, 5). Colonization among nonpregnant adults is less well studied, but vaginal and rectal colonization of healthy young and elderly adults have been reported at levels similar to those observed during pregnancy, including among males (6–9). Studies on the transmission of GBS between individuals are rare, and although transmission was found to occur between sex partners during pregnancy, multiple transmission modes may exist (6, 10). The reservoir for adult disease is poorly understood, but it is generally believed that most infections originate from the individual's commensal microbiota or from transmission from one of its close contacts (2).

There is a broad spectrum of GBS disease in adults, and these infections are responsible for substantial morbidity and mortality, particularly in individuals with chronic underlying conditions (11). The incidence of GBS infections in adults has increased in recent years, and an aging population with more comorbidities is a possible explanation (12). Diabetes is an important predisposing factor for GBS skin and soft tissue infections by contributing to immune senescence and altered integrity of anatomical barriers that promote GBS invasion, particularly among older individuals (8, 11, 13). Notwithstanding host factors, bacterial factors could also play a role in disease incidence and severity. The invasive potential of a given genetic lineage can be evaluated by comparing its prevalence among asymptomatic carriers, the reservoir for infection, and its prevalence in infection (case-carrier ratios). Such comparisons between vaginal carriage and infant disease identified the multilocus sequence type 17 (ST17) lineage as being particularly virulent (14).

Considering that GBS asymptomatically colonizes the human host and that this is potentially a prerequisite for disease, it is crucial to extend our knowledge of the colonizing bacterial population of nonpregnant adults to better devise preventive strategies against GBS infections in this group. The aim of this study was to evaluate GBS colonization among community-dwelling nonpregnant adults and to characterize their genetic diversity. Comparison of adult colonization isolates with those causing invasive disease was then performed to identify lineages with propensity to colonize different anatomical sites or with an enhanced invasive disease potential.

## RESULTS

**Study participants.** Figure 1 presents information on the enrolled participants, with 336 participants having been included in the study. There were no significant differences in gender distribution by age group. In most analyses, all participants of ≥60 years were considered a single group. Twenty-four participants (7%) reported being diabetic, mostly with type II diabetes mellitus ($n = 21$, 87.5%), including all 13 participants recruited at APDP. Diabetic participants had the same gender balance but were older ($P < 0.001$) than nondiabetic participants.

**GBS colonization by site, age, and gender.** A total of 107 participants (31.8%; 95% confidence interval [$CI_{95}$] of 27.1% to 37.0%) were colonized with GBS in ≥1 sample, with GBS being recovered at similar rates in females ($n = 67/211$; 31.8%) and males ($n = 40/125$; 32.0%). Sixty-two colonized females (93%) had a positive vaginal sample, 59 (88%) a positive urine sample, 57 (85%) a positive anorectal sample, and 4 (6%) a positive oral sample. Among colonized males, GBS was detected in 31 (78%) anorectal samples, in 23 (58%) urine samples, and in 5 (13%) oral samples. Most colonized females ($n = 46$; 69%) were positive for GBS simultaneously in the vaginal, anorectal, and urine samples. Most males were positive for GBS in the anorectal samples only ($n = 14$; 35%) or in both anorectal and urine samples ($n = 13$; 33%). Only 19 participants (10 females and 9 males) were colonized in the genital tract (vagina or urine) but not anorectally. In total, 78 participants (73%) were colonized with GBS in ≥2 samples.

Among females, GBS colonization was stable at about 25% in the childbearing-age groups (18 to 29 and 30 to 44 years) and increased to 42% in the ≥60 years group

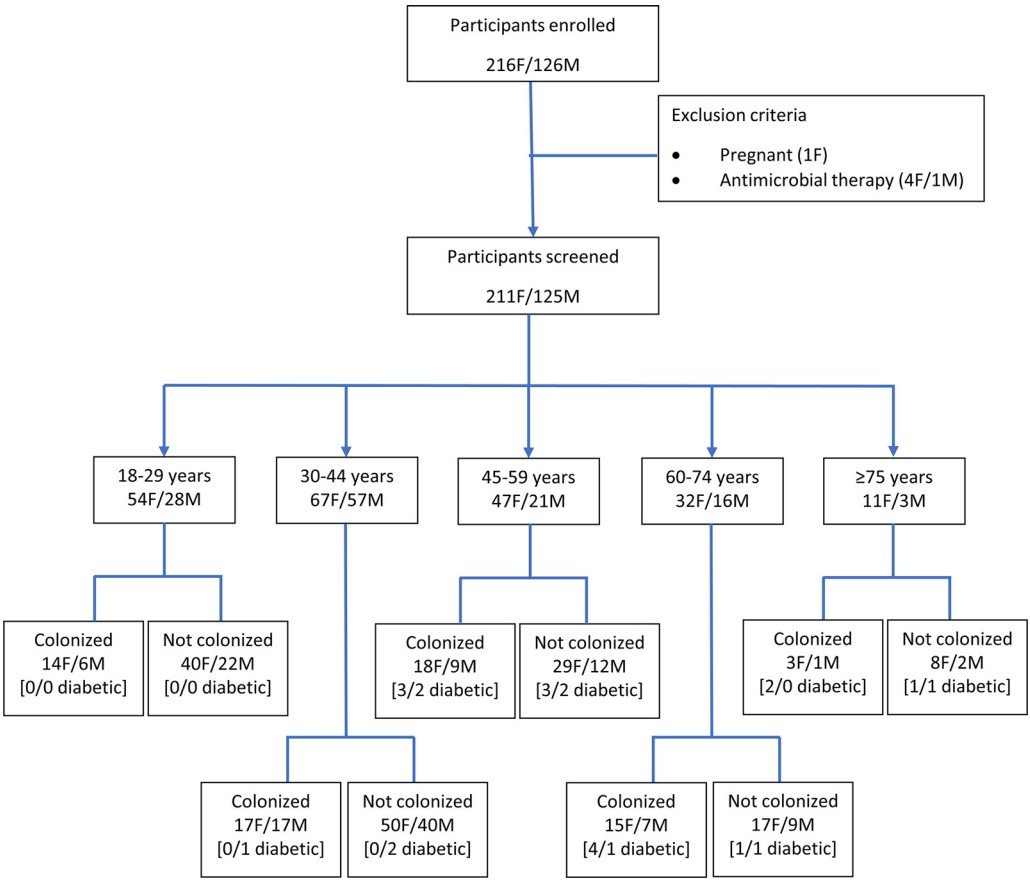

**FIG 1** Study recruitment flowchart. F, female; M, male.

($P$ = 0.040) (Fig. 1 and 2). Despite the same overall colonization rate among males and females, there were differences between age groups, but none were statistically supported after false-discovery rate (FDR) correction. Colonization of males also increased with age from 21% in the 18 to 29 years group to over 42% in the ≥60 years group, but this trend did not reach statistical significance ($P$ = 0.072). GBS colonization among diabetic participants ($n$ = 13/24; 54%) was higher than that found among nondiabetic participants ($n$ = 94/312; 30%), but when controlling for age this difference was not significant.

**Serotypes, genetic lineages, and antimicrobial resistance.** For each of the 107 GBS colonized participants, one GBS isolate per sample was characterized, for a total of 243 isolates. In 99 participants, the isolates recovered from different samples were phenotypically

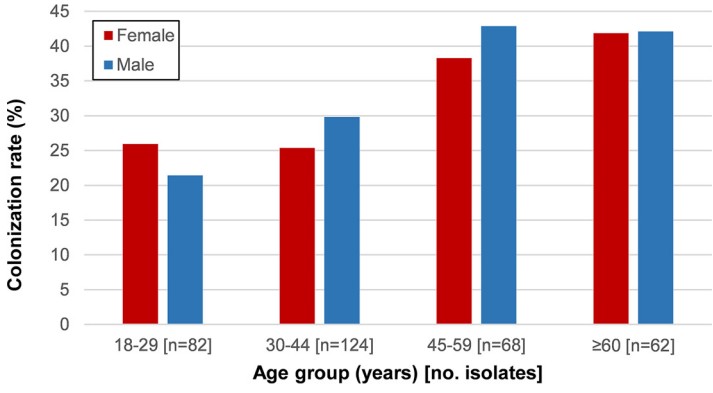

**FIG 2** GBS colonization rate by gender and age group. Red bars represent the proportion of colonized females and the blue bars that of colonized males.

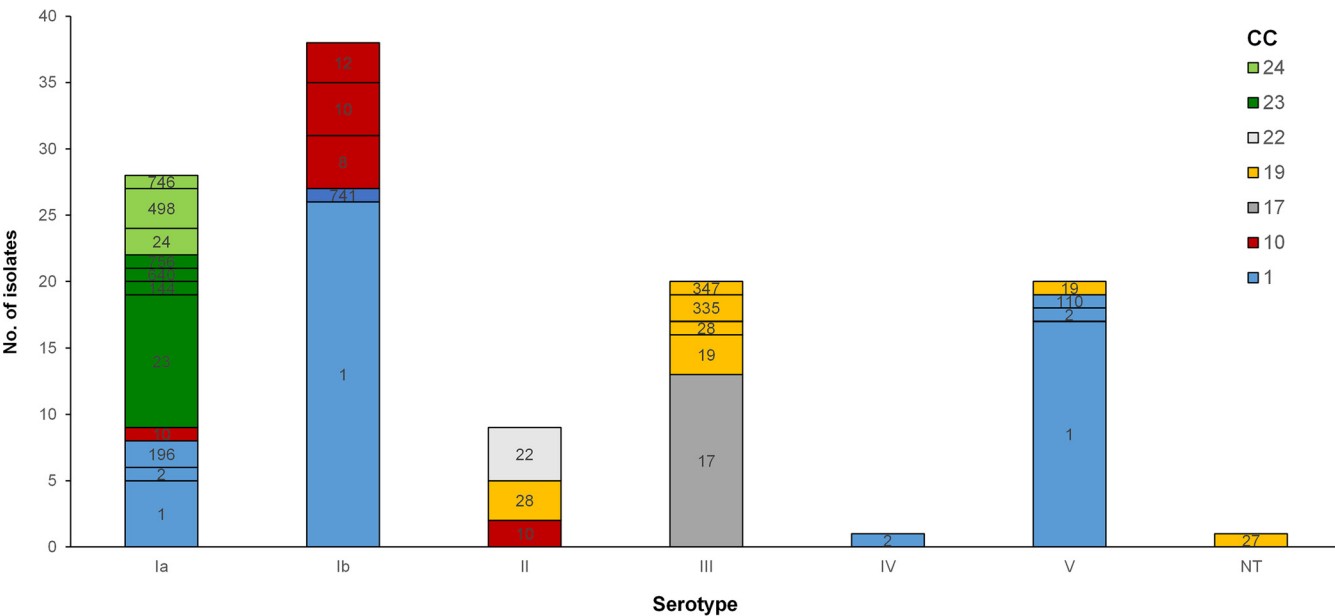

**FIG 3** Serotype distribution by MLST of GBS colonizing nonpregnant adults. Labels within bars indicate STs, colored according to their clonal complex (defined at the SLV level). ST23 and ST24 and respective SLVs were grouped into two clonal complexes—CC23 and CC24—and given different shades of green because they are DLVs.

and genotypically indistinguishable, and only one representative was used when evaluating the genetic diversity of GBS colonization. In eight cases, two distinct isolates were identified in different samples of the same participant ($n = 8/107$, 7%). In six participants, mostly female ($n = 5/6$), two different serotype/ST combinations were found in different samples; in one participant, a different serotype but the same ST were found in different samples, and in another participant two different STs, albeit of the same serotype, were found for a total of 115 nonduplicate GBS isolates. In most analyses, we used the number of colonized individuals as the denominator ($n = 107$) and considered all nonduplicate isolates ($n = 115$).

The colonizing isolates had diverse serotypes (Simpson's index of diversity [SID] = 0.780, $CI_{95} = 0.748$ to 0.813) (Fig. 3), with the most frequent, Ib ($n = 38$) and Ia ($n = 25$), accounting for over half of colonized participants. Serotype V colonization was more frequent in the ≥60 years group ($P = 0.027$, but not supported after FDR).

The characteristics of the genetic lineages found are summarized in Table 1. The isolates were distributed across 22 STs (SID = 0.807; $CI_{95} = 0.750$ to 0.879) and grouped into 7 clonal complexes (CCs) (Table 1). Three novel STs (ST741, ST746, and ST756) were identified in this study. The different CCs were similarly distributed in all age groups, except for CC10, which was found only among participants of <60 years ($P = 0.019$, but not supported after FDR), and CC1, which was overrepresented in the ≥60 years group ($P = 0.032$, also not supported after FDR). Serotype Ib/CC1/*alp*3/PI-1+PI-2a was the dominant genetic lineage colonizing nonpregnant adults, a lineage also prevalent among adult infections (15), followed by the classical serotype V/CC1, sharing the same genetic characteristics, and the infant hypervirulent serotype III/ST17 lineage. There was no association between CC and sample type.

**Antimicrobial susceptibility.** Antimicrobial resistance genotypes are presented in Table 1. All isolates were susceptible to penicillin, vancomycin, chloramphenicol, and gentamicin. Nonsusceptibility to levofloxacin was identified in 4.3% ($n = 5$) of the isolates. Tetracycline resistance was found in 89.6% of the isolates ($n = 103$), associated mostly with the *tet*(M) gene. The overall rates of erythromycin and clindamycin resistance were 41.7% ($n = 48$) and 40.9% ($n = 47$), respectively. Erythromycin resistance was overrepresented within CC1 ($P < 0.001$) and serotype Ib ($P < 0.001$), with all serotype Ib/CC1 isolates presenting the cMLS$_B$ phenotype and carrying the *erm*(B) gene. On the other hand, serotype Ia was associated with erythromycin susceptibility ($P = 0.011$). Three isolates (2.6%) presented high-level resistance to streptomycin, all carrying *aph(3')-IIIa* and *ant(6)-Ia*, two genetic

**TABLE 1** Distribution of serotypes, surface protein, pili, and antimicrobial resistance among multilocus sequence typing-based clonal complexes of group B *Streptococcus* colonizing nonpregnant adults

| Genetic lineage[a] (n) | Serotype[b]/Alp[c] gene/pili (n) | Macrolide resistance phenotype[d] (n) | Antimicrobial resistance genotypes (n) | | |
|---|---|---|---|---|---|
| | | | Macrolide(s) | Tetracycline(s) | Aminoglycoside(s) |
| CC1 (52) | | | | | |
| ST1 (46) | Ib/*alp3*/PI-1+PI-2a (26) | cMLS$_B$ (26) | *erm*B (26) | *tet*M (25) | |
| | V/*alp3*/PI-1+PI-2a (17) | cMLS$_B$ (1), iMLS$_B$ (4) | *erm*B (1), *erm*TR (4) | *tet*M (14), *tet*M+*tet*O (1) | |
| | V/*rib*/PI-1+PI-2a (1) | | | *tet*M (1) | |
| | Ia/*alp3*/PI-1+PI-2a (2) | | *erm*B (2) | *tet*M (2) | |
| ST2(3) | Ia/*eps*/PI-1+PI-2a (1) | | | *tet*M (1) | |
| | IV/*eps*/PI-1+PI-2a (1) | | | *tet*M (1) | |
| | V/*eps*/PI-1+PI-2a (1) | cMLS$_B$ (1) | *erm*B (1) | *tet*M (1) | |
| ST196(2) | Ia/*eps*/PI-1+PI-2a (2) | cMLS$_B$ (1) | *erm*B (1) | *tet*M (1), *tet*M+*tet*O (1) | *aph(3')-IIIa*+*ant(6)-Ia* (1) |
| ST741(1) | Ib/*alp3*/PI-1+PI-2a (1) | cMLS$_B$ (1) | *erm*B (1) | *tet*M (1) | |
| CC10 (14) | | | | | |
| ST8(4) | Ib/*bca*/PI-1+PI-2a (4) | | | *tet*M (4) | |
| ST10(7) | Ia/*bca*/PI-1+PI-2a (1) | | | | |
| | Ib/*bca*/PI-1+PI-2a (4) | | | *tet*M (1) | |
| | II/*bca*/PI-1+PI-2a (2) | iMLS$_B$ (2) | *erm*T (2) | *tet*M (2) | |
| ST12(3) | Ib/*bca*/PI-1+PI-2a (3) | | | *tet*M+*tet*O (2), *tet*O (1) | |
| CC17 (13) | | | | | |
| ST17(13) | III/*rib*/PI-1+PI-2b (11) | | | *tet*M (11) | |
| | III/*rib*/PI-2b (2) | cMLS$_B$ (2) | *erm*TR+*mef*E (2) | *tet*O (2) | *aph(3')-IIIa*+*ant(6)-Ia* (2) |
| CC19 (13) | | | | | |
| ST19(4) | III/*rib*/PI-1+PI-2a (3) | iMLS$_B$ (2) | *erm*TR (2) | *tet*M (2) | |
| | V/*rib*/PI-1+PI-2a (1) | | | *tet*M (1) | |
| ST27(1) | NT/*rib*/PI-1+PI-2a (1) | | | | |
| ST28(4) | II/*rib*/PI-1+PI-2a (3) | iMLS$_B$ (1) | *erm*T (1) | *tet*M (3) | |
| | III/*rib*/PI-1+PI-2a (1) | | | *tet*M (1) | |
| ST110(1) | V/*rib*/PI-2a (1) | cMLS$_B$ (1) | *erm*B (1) | *tet*O (1) | |
| ST335(2) | III/*rib*/PI-1+PI-2a (2) | iMLS$_B$ (2) | *erm*TR (2) | *tet*M (2) | |
| ST347(1) | III/*rib*/PI-1+PI-2a (1) | | | *tet*M (1) | |
| CC22 (4) | | | | | |
| ST22(4) | Ia/*bca*/PI-2a (1) | | | *tet*M (1) | |
| | II/*bca*/PI-2a (3) | cMLS$_B$ (1) | *erm*B (1) | *tet*M (1) | |
| CC23/CC24 (13/6) | | | | | |
| ST23(10) | Ia/*eps*/PI-2a (10) | cMLS$_B$ (1) | *erm*TR (1) | *tet*M (10) | |
| ST144(1) | Ia/*rib*/PI-1+PI-2a (1) | | | *tet*M (1) | |
| ST640(1) | Ia/*eps*/PI-2a (1) | M (1) | *mef*E (1) | *tet*M (1) | |
| ST756(1) | Ia/*alp2*/PI-1+PI-2a (1) | cMLS$_B$ (1) | *erm*TR (1) | | |
| ST24(2) | Ia/*bca*/PI-2a (2) | | | *tet*M (2) | |
| ST498(3) | Ia/*bca*/PI-2a (3) | | | *tet*M (3) | |
| ST746(1) | Ia/*bca*/PI-2a (1) | | | *tet*M (1) | |

[a]CC, clonal complex; ST, sequence type.
[b]NT, nontypeable.
[c]Alp, alpha/alpha-like protein.
[d]M, resistance to macrolides; MLS$_B$, resistance to macrolides, lincosamides, and streptogramins B. The prefix letter refers to the constitutive expression of this phenotype (cMLS$_B$) or inducible expression of the phenotype (iMLS$_B$).

determinants encoding aminoglycoside-modifying enzymes. Two of these isolates belong to the serotype III/CC17/PI-2b sublineage of the hypervirulent lineage, being simultaneously resistant to macrolides, lincosamides, and tetracycline, a sublineage previously reported among neonatal (16) and adult (15) invasive disease in Portugal.

**Comparison with contemporary isolates causing invasive disease in adults.** We considered all invasive isolates recovered from adults (≥18 years) identified in clinical microbiology laboratories throughout Portugal and not only from the Lisbon area from February 2013 to October 2015 (15) as contemporary to this colonization study. Most of the 268 invasive disease isolates were recovered from blood (85%, n = 229), while the remaining were recovered from other normally sterile fluids, including cerebrospinal fluid (CSF), ascitic, synovial, and pleural fluid, and aqueous humor. The diversity of the invasive disease isolates

in terms of serotype (SID = 0.779, $CI_{95}$ = 0.755 to 0.802) and ST (SID = 0.815, $CI_{95}$ = 0.773 to 0.857) was similar to that of the isolates colonizing GBS. Despite differences in serotype and CC distribution between colonization and invasive disease isolates (see Fig. S1 in the supplemental material), none were significant, even when stratifying by age. A more detailed analysis, combining all typing information available to define GBS lineages (serotype, ST, surface protein gene, and pilus islets), also did not reveal any difference between the colonizing and invasive disease-causing bacteria. In both GBS populations, the serotype Ib/CC1/*alp*3/PI-1+PI-2a was the dominant lineage, a lineage which was shown to result from recombination of a large exogenous DNA fragment leading to a capsular switching event (15). These data are consistent with a homogenous potential to cause adult invasive disease of the various GBS circulating lineages.

## DISCUSSION

The prevalence of GBS carriage in our study (31.8%; $CI_{95}$ = 27.1% to 37.0%), including both males and females, is closer to the highest reported in the two most recent studies of GBS colonization during pregnancy in Portugal—21% (17) and 35% (18)—and higher than that found in a meta-analysis of studies following similar microbiological methods (19.4%; $CI_{95}$ = 16.1% to 23.1%) (9). The increasing proportion of GBS colonized individuals with age found in our study, independent of gender, is in contrast to a French study, in which colonization rates peaked in the 41 to 54 years group but then decreased in older participants (19). An increase of colonization in older adults was also not found in the meta-analysis (9). Despite more than half of the small number of diabetic participants in our study (*n* = 24) being colonized with GBS, this was not different from the proportion of colonization among nondiabetic participants when adjusted for age. Although diabetes is associated with an increased risk of GBS disease among nonpregnant adults (13), this does not seem to be due to a higher colonization.

GBS colonization and disease appear to be restricted to a limited number of lineages, despite differences in their prevalence in time and according to geographic location. Our study shows a similar distribution of serotypes and lineages in contemporary colonizing and invasive isolates in a geographically restricted adult population, strongly suggesting that there are no GBS lineages with a particular propensity to cause invasive disease in adults, in contrast to the enhanced invasiveness of serotype III/ST17 lineage in infants (14) or the various lineages of *Streptococcus pneumoniae* showing higher invasiveness in both children and adults (20). The serotype Ib/CC1 was the dominant lineage in both carriage and invasive disease in Portugal in 2013 to 2015. Our study suggests that an increase in colonization by this capsular-transformed lineage may be behind its increased prevalence in invasive GBS disease, which was a major cause of an increase in macrolide resistance (15). Taken together, our data are consistent with an unprecedented and extremely fast dissemination of this GBS lineage in Portugal in invasive disease but also in asymptomatic colonization of adults in Portugal. The reasons for its success in Portugal and not elsewhere remain elusive. Other antimicrobial resistant lineages were also present in colonization, including the multiresistant serotype III/CC17/PI-2b sublineage of the hypervirulent ST17 lineage. The serotype distribution found in our study differed considerably from that of the 21 studies reviewed in the recent meta-analysis, where serotypes III and II were the most prevalent, but this could be due to the fact that most reviewed data were published before 1990 (9). This suggests that temporal changes in GBS colonization have occurred, supporting the need for additional studies of colonization in nonpregnant adults to better understand the relationship between colonization and disease by this important pathogen.

Our study has several limitations. While the carriage study was performed in the Greater Lisbon area, the isolates responsible for invasive disease were collected in hospitals scattered throughout Portugal. Although there may be differences in the prevalence of colonization between regions in Portugal, we have no reason to think that there would be differences in the lineages and their prevalence between regions. In fact, when stratifying the invasive isolates by geographic region, no differences could be found (data not shown). The demographic characteristics of the population recruited into the study do not mirror exactly those of the Portuguese population or the population with GBS invasive infections, in terms

of gender distribution and age structure. But again, although we found differences in carriage prevalence, we could not show differences in the characteristics of the carriage isolates across these groups, so we do not feel that this would introduce a significant bias. The characterization of a single colony from each site in our study ignored the possibility of carriage of multiple strains at the same site and may have underestimated the prevalence of certain serotypes and lineages, but this could have led to overlooking the identification only of lineages particularly associated with asymptomatic carriage and not of particularly virulent lineages. While the carriage isolates were recovered in two very limited periods, the invasive isolates were recovered over an overlapping but longer period. This decision was guided by a greater statistical efficiency in the comparisons, and given that the GBS population was found to be clonally stable in that period (15), this should not affect our conclusions. We relied on self-reporting for the identification of diabetic participants in the community. This could have led to an underestimation of the true prevalence of diabetes among the participants, particularly because of a potentially significant prevalence of undiagnosed diabetes. The increasing incidence of diabetes with age could have therefore acted as a confounder for the increased GBS colonization with age observed in the study. Although we recruited 336 participants, of which 107 were colonized with GBS, and analyzed 268 invasive isolates, it is possible that a greater statistical efficiency gained from analyzing a larger number of colonization isolates would detect finer differences between the two populations. However, this should affect mostly minority lineages in both colonization and disease and should not change the conclusions relative to the lineages representing most of the isolates. Finally, we cannot exclude that more subtle differences between the invasive and colonization isolates exist that could not be detected by the typing methods employed. A future genomic analysis of the isolates will further explore this. However, multilocus sequence typing alone was able to identify ST17 as a particularly virulent lineage in neonatal infections, while the conjunction of different methods employed here was not able to identify any such lineage in adult infections.

We found that GBS colonization in adults increased with age, potentially explaining the higher incidence of GBS infections in older individuals. Colonization with strains indistinguishable by the methods used in different sites was the norm, conforming with no niche specialization of the different circulating GBS lineages and with transfer of GBS between anatomical sites in the same individual. The lineages and their prevalence found in carriage are not significantly different from those causing invasive disease in adults, consistent with GBS acting as an opportunistic pathogen with no particularly virulent lineages for adult infections.

## MATERIALS AND METHODS

**Recruitment of participants and sample collection.** Nonpregnant adults (≥18 years) living in the community (i.e., not residing in elderly homes or long-term-care facilities), including students attending the Faculdade de Medicina of Universidade de Lisboa (FMUL), personnel from FMUL and Instituto de Medicina Molecular João Lobo Antunes, and their community contacts in a snowball sampling approach, as well as patients attending a diabetes follow-up clinic, were invited to participate in the study. Our study was performed in two periods: February to March 2013 and September 2014 to October 2015 in the same institutions and with the same protocol.

All participants provided informed consent. Detailed information on the self-collection of samples and a collection package were supplied to each participant, and the packages' contents were numbered using a coded system to protect the identity of the participant. Participants provided a self-collected initial-void urine sample, an oral (cheek) swab, and an anorectal swab, with female participants also providing a lower vaginal swab, and completed a brief anonymous questionnaire including information on age, gender, diabetes mellitus status and type, pregnancy status, and antimicrobial consumption in the previous 7 days. Participants were excluded from further study if reporting either being pregnant or having taken an antibiotic in the week prior to sample collection.

**Sample processing.** Collection packages were recovered from participants the same day the samples were taken and processed within 24 h after collection. Swabs and 100 $\mu$L of urine sediment were inoculated into Todd-Hewitt broth supplemented with colistin (10 $\mu$g/mL) and nalidixic acid (15 $\mu$g/mL) and incubated overnight at 37°C under normal atmosphere. A 10-$\mu$L loopful of each culture was then inoculated onto chromogenic agar (CHROMID Strepto B, bioMérieux, Marcy L'Étoile, France) that presumptively identifies GBS by a color change, including nonhemolytic variants. Following overnight incubation at 37°C under normal atmosphere, one suspected colony from each culture was subcultured into Trypticase-soy agar supplemented with 5% sheep blood and GBS identification was confirmed by Gram staining, by colony morphology, and serologically with a streptococcal grouping latex agglutination test (Slidex Strepto B; bioMérieux, Marcy L'Étoile, France).

**Isolate characterization.** All isolates were serotyped by a slide agglutination assay with IMMULEX STREP-B kit (Statens Serum Institute, Copenhagen, Denmark) according to the manufacturer's instructions. Susceptibility testing to penicillin G, erythromycin, clindamycin, vancomycin, chloramphenicol, levofloxacin,

and tetracycline was performed by disc diffusion according to the Clinical and Laboratory Standards Institute (CLSI) guidelines (21). An additional screening test for penicillin reduced susceptibility was performed using a combination of ceftibuten (30 $\mu$g), oxacillin (10 $\mu$g), and ceftizoxime (30 $\mu$g) disks (22). High-level aminoglycoside resistance (HLAR) was tested according to CLSI methods and interpretative criteria for *Enterococcus* species (21) but using Mueller-Hinton supplemented with sheep blood. Macrolide-, lincosamide-, tetracycline-, and aminoglycoside-resistant isolates were examined for the presence of resistance determinants (15).

**MLST and surface protein gene and pili profiling.** Multilocus sequence typing (MLST) was performed as described previously (23), and alleles and sequence types (STs) not previously described were deposited in the *S. agalactiae* MLST database (http://pubmlst.org/sagalactiae). The goeBURST algorithm implemented in PHYLOViZ (24) was used to establish relationships between STs. Clonal complexes (CCs) were defined at the single locus-variant (SLV) level. The GBS alpha-C (*bca*), alpha-like protein genes (*eps*, *rib*, *alp2*, *alp3*, and *alp4*), and pilus islands were detected by PCR (25).

**Typing analysis and statistics.** Simpson's index of diversity (SID) and 95% confidence intervals (CI$_{95}$) were used to estimate diversity (www.comparingpartitions.info) (26). Differences were evaluated by Fisher exact test with false-discovery rate (FDR) correction for multiple testing. The Cochran-Armitage test was used for trends. A *P* value of <0.05 was considered significant for all tests.

**Ethical statement.** The project was submitted and approved by the institutional review boards of Centro Académico de Medicina de Lisboa (CAML) and of Associação Protectora dos Diabéticos de Portugal (APDP).

## SUPPLEMENTAL MATERIAL

Supplemental material is available online only.

**SUPPLEMENTAL FILE 1**, PDF file, 0.1 MB.

## ACKNOWLEDGMENTS

We thank Joana Lopes for technical support.

Conceptualization, E.R.M. and M.R.; methodology, J.M.-C., E.R.M., and M.R.; formal analysis, E.R.M. and M.R.; investigation, E.R.M., D.N.d.O., and A.L.M.C.; data curation, E.R.M.; writing—original draft preparation, E.R.M.; writing—review and editing, J.M.C., E.R.M., and M.R.; project administration, J.M.C. and M.R. All authors have read and agreed to the published version of the manuscript.

ERM was supported by Fundação para a Ciência e a Tecnologia (SFRH/BPD/80038/2011 and DL57/2016/CP1451/CT0009).

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
