## [Reviewer comments · Microbiology Spectrum]

Microbiology Spectrum

Characteristics of *Streptococcus agalactiae* colonizing non-pregnant adults support the opportunistic nature of invasive infections

Elisabete Martins, Dulce Nascimento do Ó, Ana Marques-Costa, Jose Melo-Cristino, and Mario Ramirez

Corresponding Author(s): Mario Ramirez, Faculdade de Medicina, Universidade de Lisboa

Review Timeline:

Submission Date:	March 24, 2022
Editorial Decision:	April 19, 2022
Revision Received:	April 29, 2022
Accepted:	May 4, 2022

Editor: Jennifer Auchtung

Reviewer(s): The reviewers have opted to remain anonymous.

Transaction Report:

DOI: <https://doi.org/10.1128/spectrum.01082-22>

April 19, 2022

Dr. Mario Ramirez
Faculdade de Medicina, Universidade de Lisboa
Instituto de Medicina Molecular, Instituto de Microbiologia
Av. Prof. Egas Moniz
Lisboa 1649-028
Portugal

Re: Spectrum01082-22 (Characteristics of Streptococcus agalactiae colonizing non-pregnant adults support the opportunistic nature of invasive infections)

Dear Dr. Mario Ramirez:

Thank you for submitting your manuscript to Microbiology Spectrum. We've received two reviews of your work which are generally favorable, although both reviewers highlighted information that should be included (more details of study periods/sample collection strategy and/or commented on (potential limitations on statistical analyses due to small sample size) that should be addressed in your revisions. When submitting the revised version of your paper, please provide (1) point-by-point responses to the issues raised by the reviewers as file type "Response to Reviewers," not in your cover letter, and (2) a PDF file that indicates the changes from the original submission (by highlighting or underlining the changes) as file type "Marked Up Manuscript - For Review Only". Please use this link to submit your revised manuscript - we strongly recommend that you submit your paper within the next 60 days or reach out to me. Detailed instructions on submitting your revised paper are below.

Link Not Available

Sincerely,

Jennifer Auchtung

Journals Department
Reviewer comments:

Reviewer #1 (Comments for the Author):

The authors describe prevalence and lineages of colonizing GBS among non-pregnant adults and compare to contemporary isolates from invasive disease cases in Portugal. The data show increase rates of colonization with age in the adult population with highest rates in the ≥ 60 yr group. Predominant serotype and lineages are provided and compared to those from invasive disease showing similarities and no particular clone associated with disease in the non-pregnant adults. While the data

presented is from 2013-2015 it does provide regional information on GBS serotypes and lineages and provides data that contributes to our understanding of strains colonizing non-pregnant adults.

Comments

I might have missed this presented in the manuscript but were any comparisons done between the 2 study periods? Were the same facilities/institutions used in both study periods for sample collection. Was there a reason for 1st study period being conducted in Feb/Mar and 2nd in Sept/Oct?

Page 14, lines 252-253. Can you include in the Methods section some additional information on the contemporary invasive GBS isolates included in the study. There is very little detail included. You mention timeline of Feb 2013-Oct 2015 isolates and assume you were only looking at adults ≥ 18 yrs to be in-line with the non-invasive study? You did not restrict this sampling from only Lisbon but included isolates from entire country? What types of specimens were these invasive isolates from? Blood, CSF?

Page 15, lines 294-297. The data that you present does not support this statement that a specific lineage disseminated rapidly? I don't see any time series data showing temporal changes? Please clarify or rephrase.

Page 16, line 333. "has" should be "as"

Page 17, line 338. I think "undistinguishable" should be "indistinguishable"? Please correct throughout the manuscript.

Reviewer #2 (Comments for the Author):

The article by Martins et al. describes the epidemiology of Group B Streptococcus (GBS) colonization in 336 non-pregnant adults in Portugal as well as the phenotypic (capsular serotype and antibiotic resistance) and the genetic features (sequence type, clonal complex, presence of the alphaC, alpha-like and pili encoding genes) of the GBS colonizing isolates. Last, the serotypes and the genetic lineage of the GBS strains colonizing 107 participants were compared to those of 268 contemporary and previously described invasive isolates and were found to be similar, leading to the conclusion that GBS colonizing non-pregnant adults supports the opportunistic nature of invasive infections.

The introduction and discussion sections are well documented, the methods and results section are clear and concise and the conclusions are supported by the data. The main weakness is the limited number of patients included which might decrease the statistical power of the analyses, a point that should be further discussed.

Overall the paper reads well and I only have minor remarks:

1. Please add a legend to all the figures (a legend is missing for figures 1 and 2).
2. Line 58-59: the sentence "This means that prophylactic approaches..." is unclear to me. Do you mean that prophylactic approaches will not impact GBS colonization in adults or that they should not target particular lineages in order to protect adults from GBS invasive disease ?

Staff Comments:

Preparing Revision Guidelines

Please return the manuscript within 60 days; if you cannot complete the modification within this time period, please contact me. If you do not wish to modify the manuscript and prefer to submit it to another journal, please notify me of your decision immediately so that the manuscript may be formally withdrawn from consideration by Microbiology Spectrum.

Reviewer #1

1) I might have missed this presented in the manuscript but were any comparisons done between the 2 study periods? Were the same facilities/institutions used in both study periods for sample collection. Was there a reason for 1st study period being conducted in Feb/Mar and 2nd in Sept/Oct?

The study was started in February 2013 and our initial intention was to complete it by February 2015. However, for personal reasons, the lead investigator had to take leave from work from April 2013 to August 2014. The study was therefore interrupted for this time. The same facilities and institutions were used during both study periods and the study protocol was also the same. Given the short 2-month period of the initial sample collection, only 13 colonized individuals were identified during this period. This small number of bacterial isolates shows no notable differences to those of the longer collection period. The rate of colonization was also not different between the two periods. Given the personal nature underlying the interruption between the two collection periods and the fact that no notable differences were identified between the two periods, we did not find that discussing these points in the text would add any relevant information. However, following the reviewer's comment and to clarify in the text that the same institutions and protocol were used in both periods we have added this information in the text that now reads: "Our study was performed in two periods: February-March 2013 and September 2014-October 2015 in the same institutions and with the same protocol."

2) Page 14, lines 252-253. Can you include in the Methods section some additional information on the contemporary invasive GBS isolates included in the study. There is very little detail included. You mention timeline of Feb 2013-Oct 2015 isolates and assume you were only looking at adults ≥ 18 yrs to be in-line with the non-invasive study? You did not restrict this sampling from only Lisbon but included isolates from entire country? What types of specimens were these invasive isolates from? Blood, CSF?

We thank the reviewer for pointing out an aspect that would deserve further clarification in the text. Indeed, we included isolates from all over Portugal, previously already discussed as a possible limitation in the paper, and included only adult samples of normally sterile products. To address the issue raised by the reviewer and to offer the additional information requested, we have now altered the text to read: "We considered all invasive isolates recovered from adults (≥ 18 yrs) identified in clinical microbiology laboratories throughout Portugal and not only from the Lisbon area, from February 2013 to October 2015 (17) as contemporary to this colonization study. Most of the 268 invasive disease isolates were recovered from blood (85%, $n=229$) while the remaining were recovered from other normally sterile fluids including CSF, ascitic, synovial and pleural fluid, and aqueous humor."

3) Page 15, lines 294-297. The data that you present does not support this statement that a specific lineage disseminated rapidly? I don't see any time series data showing temporal changes? Please clarify or rephrase.

Indeed, the reviewer is right in that we do not perform any temporal analysis. Such an analysis is not possible with our study design and the relatively short collection periods. However, our previous published studies of adult invasive disease have shown that there was a large increase of serotype 1b in adult invasive disease, which was accompanied by the detection of this capsular transformed lineage (see reference 17). This occurred mostly from 2009 to 2013. By the time that our colonization study was started, this serotype 1b lineage was already abundantly found in invasive disease, while we would have expected its increase in colonization to have preceded or at most accompanied its isolation from invasive disease cases. Given that we do not find any evidence that there are lineages particularly associated with colonization or infection, we hypothesize that the rise of this lineage in invasive disease reflected a rise also in colonization. Our data supports this since we found similarly high proportions of this lineage in colonization and invasive infection at a time in which this lineage was already well established in invasive disease. To address the reviewer's concern and to make this point clearer in the text we have rephrased the relevant sentences that now read: "Our study suggests that an increase in colonization by this capsular-transformed lineage may be behind its increased prevalence in invasive GBS disease, which was a major cause of an increase in macrolide resistance (17). Taken together our data are consistent with an unprecedented and extremely fast dissemination of this GBS lineage in Portugal in invasive disease but also in asymptotically colonization of adults in Portugal."

4) Page 16, line 333. "has" should be "as"

We thank the reviewer for pointing out this mistake which we have now corrected.

5) Page 17, line 338. I think "undistinguishable" should be "indistinguishable"? Please correct throughout the manuscript.

We have altered the spelling to "indistinguishable" in the two instances the word was used in the text as requested by the reviewer.

Reviewer #2

1) The introduction and discussion sections are well documented, the methods and results section are clear and concise and the conclusions are supported by the data. The main weakness is the limited number of patients included which might decrease the statistical power of the analyses, a point that should be further discussed.

We thank the reviewer for the positive appraisal of our work. We realize that negative results always cause us to question if our study was not sufficiently powered to detect any existing differences. Naturally, having more isolates is always helpful, but we do not believe that our analysis is compromised by this. We have analyzed 107 colonization isolates and 268 invasive isolates (a total of 375 isolates). We could not increase the number of invasive isolates since these reflect the incidence of GBS invasive infections in Portugal, but an additional effort could have been done to recruit more participants into the colonization study beyond the 336 included here. However, analyses of smaller numbers of isolates was enough to identify associations between serotypes and early-onset (EOD) or late-onset (LOD) neonatal disease, and the hypervirulent ST17 lineage. As an example, our own prior work (Martins et al. 2007. *J Clin Microbiol* 45:3224–3229) analyzing 333 isolates (269 from vaginal colonization and 64 from invasive infections, a more imbalanced sample than the one in our current paper) was able to identify associations of serotype Ib with EOD (only 42 isolates in EOD) and serotype III in LOD (only 22 isolates in LOD). In the same paper we identified the high invasive disease potential of ST17, which was represented by 38 isolates distributed in infection and colonization. These numbers are like the ones we found in the current paper, so we believe that these are not responsible for not having identified a particularly virulent lineage for infection in adults if such a lineage would exist. Naturally, having more colonization isolates could allow us to probe more subtle differences between colonization isolates and those causing infection and, particularly, to better determine the invasive potential of low prevalence lineages. However, the general conclusion that there are no major lineages, responsible for significant numbers of infections, associated to invasive disease in adults would still stand. To address the reviewer's concern and to alert the reader to the possibility that a larger number of isolates could detect more subtle differences, we have added the following sentences to the discussion: "Although we recruited 336 participants, of which 107 were colonized with GBS, and analyzed 268 invasive isolates, it is possible that a greater statistical efficiency gained from analyzing a larger number of colonization isolates would detect finer differences between the two populations. However, this should mostly affect minority lineages in both colonization and disease and should not change the conclusions relative to the lineages representing most of the isolates."

2) Please add a legend to all the figures (a legend is missing for figures 1 and 2)

We thank the reviewer for pointing to the missing information. We have now added legends to both figures where they were missing.

3) Line 58-59: the sentence "This means that prophylactic approaches..." is unclear to me. Do you mean that prophylactic approaches will not impact GBS colonization in adults or that they should not target particular lineages in order to protect adults from GBS invasive disease?

Our intention was to say that prophylactic approaches targeting only a limited number of lineages would have a limited impact in infection since, if all lineages have the same invasive disease potential, the elimination of a given lineage would be expected to lead to its prompt substitution by another. To address the reviewer's concern and to better convey our intended meaning we have altered the sentence to read: "This means that any prophylactic approaches targeting colonization by particular lineages is expected to have a limited impact on GBS disease in adults."

May 2, 2022

Dr. Mario Ramirez
Faculdade de Medicina, Universidade de Lisboa
Instituto de Medicina Molecular, Instituto de Microbiologia
Av. Prof. Egas Moniz
Lisboa 1649-028
Portugal

Re: Spectrum01082-22R1 (Characteristics of Streptococcus agalactiae colonizing non-pregnant adults support the opportunistic nature of invasive infections)

Dear Dr. Mario Ramirez:

Your manuscript has been accepted, and I am forwarding it to the ASM Journals Department for publication. You will be notified when your proofs are ready to be viewed.

Sincerely,

Jennifer Auchtung
Editor, Microbiology Spectrum
